# The Impact of Meat Consumption on Human Health, the Environment and Animal Welfare: Perceptions and Knowledge of Pre-Service Teachers

**António Almeida** [1,2,*] , **Joana Torres** [3,4] and **Isilda Rodrigues** [5,6]

1 Instituto Politécnico de Lisboa, 1549-020 Lisboa, Portugal
2 CICS.NOVA, Universidade Nova de Lisboa, 1070-312 Lisboa, Portugal
3 Instituto de Ciências da Terra (ICT—Polo Porto), Universidade do Porto, 4169-007 Porto, Portugal; joana.torres@iees.pt
4 Escola Superior de Educação de Fafe (ESEF), Instituto Europeu de Estudos Superiores (IEES), 4824-909 Fafe, Portugal
5 School of Human and Social Sciences, University of Trás-os-Montes and Alto Douro (UTAD), 5000-801 Vila Real, Portugal; isilda@utad.pt
6 Centre for Research and Intervention in Education (CIIE), Faculty of Psychology and Education Sciences, University of Porto, 4200-135 Porto, Portugal
* Correspondence: aalmeida@eselx.ipl.pt

**Abstract:** Although meat was considered fundamental for human health in the past, in recent decades, it has come to be considered a poison. The present study involved 197 pre-service primary teachers from two Portuguese institutions: one in an urban context and the other in a smaller city. It aimed to ascertain the main diet of the participants and their perceptions and knowledge about the impact of meat consumption on human health, the environment and animal welfare. It also aimed to identify which type of meat was considered better and worse for health, how important meat consumption is for adults and children, and to determine whether the participants would consider it important to reduce meat consumption in their diets and the reasons for doing so. The study uses mainly a methodological quantitative approach, and a questionnaire was designed and applied. The participants showed their preference for an omnivorous diet with no restrictions, considering this diet the best for human health, but the worst for the environment and animal welfare. Although meat consumption was considered important for human health, most participants considered that it would be important to consume less meat. From the answers given for the open questions, it was concluded that some respondents had difficulties in justifying their answers, revealing a certain lack of knowledge about the subject, which has important implications for teacher training courses.

**Keywords:** meat consumption; human health; environment; animal welfare; teacher training courses

## 1. Introduction

The widespread consumption of meat in the human diet is a reality only a few decades old in most countries. In Portugal, until the 1960s, the main diet was predominantly vegetarian, similar to what was the case in other countries in southern Europe [1]. Graça also states that the consumption of beef was practically absent in the diet of the Portuguese, since cattle played a central role in agricultural work [1]. However, over a few decades, this scenario changed, mainly due to economic and cultural factors. The April 25 Revolution in 1974 was a turning point in Portuguese society, bringing many transformations in the life of the Portuguese, including their diet [2]. The restoration of democracy led to the economic, political and cultural openness of the country, with a progressive improvement in the welfare of its citizens. More products became available, restaurants were no longer just for the rich. People became familiar with the diet options of other countries, and meat consumption, especially beef, started to grow, following the trend of other developed or

developing countries, where there was an increase in the average individual income. Therefore, meat consumption became an indicator of wellbeing and economic development [3–6]. This openness of the Portuguese society also brought less positive aspects, such as the increase in fast food consumption. In this respect, the first McDonald's restaurant opened in Lisbon in 1991, and this and other similar fast-food chains target the young, trying to mold their taste while they are most susceptible. In fact, meat consumption, including red meat and fast food, became particularly popular among the younger generation, including those attending higher education [7,8]. This trend, popular amongst university students, appears as a consequence of the timetables and academic tasks in which they are involved, pushing them to choose quick, cheap and low-quality meals with a high caloric value, and few fruits and vegetables [8,9].

Globally, the increase in meat consumption in the world was due to social, cultural and economic factors, but the recognition of its nutritional value is without any doubt also important. "Meat has long been considered the fuel of our body and the best source of energy for humans (p. 1)" [10]. In fact, it is a rich source of protein, providing all essential amino acids and ample amounts of micronutrients: iron, zinc, calcium, folate and selenium, and vitamins (B6, B12, D) and omega-3 polyunsatured fatty acids, some of them essential to reduce nutritional deficiencies and with potential anticancer properties or capable of preventing other diseases [11,12]. In addition, many of these essential nutrients are difficult to obtain in adequate proportions in other foods, e.g., zinc and even iron [13]. In food, iron appears in heme and non-heme forms. The heme form is in different types of meat and seafood, and is better absorbed by our body than the non-heme form; the non-heme form appears in vegetables and legumes, and needs the presence of heme-iron and vitamin C to be better absorbed [14]. Other nutrients, such as vitamin B12, are only present in foods of animal origin and provide adequate support for building and maintaining muscle tissue [13].Therefore, the benefits associated with meat consumption were perhaps the most highlighted, at least in an initial phase in which its production and consumption started to grow. Consequently, the price of meat today has reached the lowest in decades when compared to the average income of the people in most countries [15], also due to the support given to the sector by the state.

However, the perception of meat as a good food has started to change, at least in the last two decades, being considered a poison capable of destroying the human body [10]. The discussion about its harmful effects began to include two more important dimensions: the environment and animal welfare. In fact, animal production and meat consumption, besides their impact on human health, affect the environment quality in different ways and raise ethical questions regarding the way animals are raised and treated.

In terms of human health, meat is susceptible to contamination during production, processing, storage, transport and marketing, that is to say, at every stage of the process, as a result of the degradation of protein and other nutrients caused by the enzymes present in animal tissues and by microbial activity [5]. As a result of its production, meat may contain antibiotics, hormones and chemicals in general. In the case of antibiotics, administered to animals to keep them healthier and more productive, there is evidence that their use involves risks for the consumer, as it generates the emergence of resistant microorganisms that, once in humans, can be difficult to destroy [16]. A variety of diseases or infections can be transmitted to humans by animals (zoonoses): tuberculosis, brucellosis, cysticercosis and food poisoning. Even so, the European Union approved legislation to reduce the environmental impact of animal production, improve animal welfare conditions and reduce veterinary drugs which can cause human health problems [17]. However, the global meat business is complex and involves meat imports from countries that do not always follow the European regulations.

Even more, there is an increase in research that shows a connection between meat consumption and certain chronic diseases such as obesity, type 2 diabetes, coronary heart disease, stroke, metabolic syndrome, colorectal cancer and mortality in general [18]. More precisely, The World Health Organization (WHO) [19] claims, as some epidemiological

studies suggest [20,21], that processed meat (the one that is salted, smoked or dried) is carcinogenic and red meat is possibly carcinogenic. The WHO also advises that less than ten percent of total energy consumption should originate from saturated fat, precisely the type present in different kinds of meat. It is also claimed that though the risk is small, it can grow, since the worldwide consumption of meat is increasing, especially in low- and middle-income countries, precisely those with an increasing population. Although some health agencies already advise limiting the intake of meat, these recommendations are mainly due to the possibility of reducing the risk of other diseases. In fact, several studies suggest that a high-fat diet, not meat consumption per se, is responsible for various types of cancer [11]. This means that the dangers must be related to the body mass index (BMI), which allows for an assessment of overweight and obesity. The incidence of the latter shows a decrease in a diet rich in fruits and vegetables and increase when associated with other harmful kinds of consumption, such as alcohol. Even so, several studies have verified that a high consumption of meat is related to an increase in the BMI [22], since it is frequently associated with a decrease in the consumption of other foods such as fish, vegetables and whole grains [23]. Very well cooked meat also increases the risks reported [13]. Consequently, poultry has come to be seen as a healthier alternative to red and processed meat due to its nutritional qualities (good source of protein and low fat content), used by agri-food firms as a food against obesity [24]. However, there has been little research into the impact of poultry consumption on health. Nevertheless, a study involving nearly half a million respondents in the UK showed that poultry meat intake was associated with gastro-esophageal reflux disease, gastritis and duodenitis, gallbladder disease and diabetes [25]. Meanwhile, a plant-based diet provides a wide range of antioxidants, which prevent some diseases such as certain types of cancer, and is associated with a lower incidence of obesity and overweight [26].

As mentioned before, a meat-based diet also has a great impact on the environment, greater than a plant-based diet. The appetite for more animal food has encouraged animal production, which has been responsible for a number of factors: deforestation for pastures and the increased land for agriculture, since crops are necessary to feed animals; decrease in biodiversity; pollution and greenhouse gas emissions from the different industrial and transport processes; and increase in the consumption of freshwater [6,27,28]. Even so, different types of animal production can have different impacts, e.g., cattle consume three times more water than poultry [15], which highlights this type of meat for reasons other than human health.

Food production is responsible for three main greenhouse gases: carbon dioxide ($CO_2$), nitrous oxide ($N_2O$) and methane ($CH_4$), and their cumulative presence in the atmosphere is complex in terms of the weight of each in global warming. Carbon dioxide has less warning potential than other gases, but a longer lifetime in the atmosphere [29]. According to Mulvarey and Robbins [30], animal production is responsible for two thirds of the human release of nitrous oxide, a gas with 296 more potential in global warming than carbon dioxide. Methane is another gas related to meat production, since the ruminants (goats, sheep and cattle) deliver it through belching due to enteric fermentation and, in a smaller percentage, through the intestine. Another main source of methane emission is manure decomposition [31]. Methane is also harmful to global warming, being 25 times more efficient at absorbing heat than carbon dioxide, despite a shorter lifetime in the atmosphere [32].

Even so, not everything is bad concerning the impact of animal production, as it provides income, food, employment and nutrients [33]. Cattle production can maintain rural areas, fixing population in depopulated areas of the territory, with an important economic, social and ecological impact. In Portugal, the abandonment of agricultural lands and their substitution by monoculture tree plantations have led to an increase in large forest fires and loss of biodiversity [34]. If cattle can promote soil erosion in arid places, it can also be an option in lands where crop production is inefficient [29]; in temperate climates, the movement of animals helps to bury seeds and dead matter in meadows which

are not ploughed. Consequently, the soil has become more aggregated and with a higher capacity for retaining water. Manure is a natural fertilizer for soil, providing nutrients such as nitrogen, phosphorus and potassium, helping plant growth, which also reduces erosion. However, the number of animals must be balanced, since the excess of these nutrients can impact negatively on groundwater and aquatic ecosystems [35–37]. Finally, grazing stimulates the underground parts of plants to grow, helping to increase carbon fixation. Therefore, good management of grasslands can mitigate climate change, but possibly without a great impact [15].

However, it is wrong to assume that a diet including meat is the only one that can have environmental impacts. According to Moskin, Plumer, Liebermen and Wengart [38], all food options have an impact, and it is equally important to know the origin of the food, the production processes and the weather conditions where it was produced, and the means of transport used to reach the consumer. Nonetheless, these authors recognize that a vegetarian or vegan diet can reduce the emission of greenhouse gases, but that a reductionist meat diet can have a similar effect. Thus, as a result of the multiple impacts of the production and consumption of food of animal origin, there have been several recommendations to reduce this type of food in the human diet [1,39]. In the case of the UK Climate Change Committee [39], the advice is a reduction of 20% in meat and dairy by 2030, helping the consumers to an informed shift for other plant-based sources of protein. This diet change can help environmental changes such as land release for woodland, peatland restoration and energy crops, minimizing climate change and the loss of biodiversity.

The need for this change was found in a survey with more than 4000 members of slow food associations from 27 countries, including Portugal: 70% of the respondents agreed that it is necessary to reduce meat consumption, since it is bad both for health and the environment. At the same time, almost 60% agreed that animals should have a just treatment and their natural behaviors be respected, which can be interpreted as a need to improve animal production conditions, but not necessarily an agreement with a diet without meat [40]. Even so, these findings involved participants from a movement which aims to change the way food is grown, produced and distributed, to build a fairer and sustainable food system and, consequently, not from a sample representative of the entire population.

In the Portuguese context economic reasons seem to be by far the most relevant factor for changes in food consumption, and have led, in periods of leaner economic times, to the replacement of red meat by poultry. However, a few other changes in the pattern of consumption have been occurring for non-economic reasons related to health, environment and the protection of animal rights, as is the case of a small decline in milk consumption [1].

It seems clear that some consumer segments are willing to pay more for food if animal welfare is considered, independently of the nutritional value of each food [41].

Singer [42] was one of the pioneers to state that there is no aspect of animal husbandry that is safe from the incursions of technology and the pressure exerted to intensify production; the production business follows a competitive commercial approach which aims to turn a quick and maximum profit, with total disregard for the dignity of animals. The horrors described by Singer, associated precisely with the use of technology, are practiced in almost all developed countries, and affect the normal development of animals, depriving them of adequate space and taking part in a community of animals of different ages, as would happen in their natural conditions, in violation of the most innate patterns and needs that their ancestors manifested in the wild. Perhaps less discussed are the genetic processes of the selection of animals aimed to choose economically favorable traits, which are often responsible for the mutilation of their welfare [43].

However, the European Union has published in the last few decades legislation to improve animal welfare in different situations, including regulations on how animals should be raised, kept and slaughtered [44]. Nonetheless, Bekoff and Pierce [45] are less optimistic about a real global change in the way animals are treated, because, in the case of food, the death of animals for consumption is still increasing.

*The Present Study*

This study involved pre-service teachers attending two higher education institutions: one in the Lisbon metropolitan area (UC-Urban Context) and the other in a smaller city in the interior north of the country, influenced by the surrounding rural context (RC—Rural Context). Its aims were the following:

(1) to identify the diet of the pre-service teachers from both institutions;
(2) to check their perception and knowledge about the impact of different diets on human health, environment and animal welfare;
(3) to assess the importance they attach to meat in the human diet, and which types of meats are healthier;
(4) to compare the results between the participants of the two institutions (urban context and rural context);
(5) to discuss the impact of food choices and diets on training courses, based on the results obtained and due to the recognized importance of the subject in education.

## 2. Methodology

The study is exploratory research that endeavored to investigate an issue not previously studied in depth, at least in Portugal. It mainly used a quantitative approach with recourse to descriptive and inferential statistics to assess the results of the closed questions of the questionnaire. Content analysis was implemented in the treatment of open questions.

### 2.1. Sample

The present study involved 197 pre-service teachers attending primary education training courses at two higher education institutions. The majority of the participants were females (179–90.9%), with 129 from the UC (Urban Context) and 68 from the RC (Rural Context), following the trend of a smaller number of males in the teaching courses in Portugal, even more pronounced among those that prepare for the first years of schooling. Participants from both institutions had a similar age average (21.7 years with an SD of 4.17 in the UC and 22.2 years with an SD of 6.93 in the RC). The unbalanced number of participants from the two institutions is due to the fact that the urban institution has more vacancies and attracts more students every school year, since it is in a more populated area.

### 2.2. Instrument and Procedures

Based on the aims of the study already presented, a questionnaire with closed and open questions, was designed. Besides the initial data about the participants (age, sex, course, institution), the questionnaire had 5 sections. The questions aimed to address the following topics: (i) to identify the most common diet of the participants among the following: lacto-ovo vegetarian, omnivore without any restrictions, vegan, lacto-ovo vegetarian and seafood, omnivore with red meat restriction (Section 1); (ii) to identify which of these diets were considered the best and the worst for human health, the environment and animal welfare, and why (Sections 2–4); (iii) to find out which types of meat were considered more and less healthy and why; iv) to identify the importance given to meat in human health in general and for children in particular; (v) to gauge the readiness of the participants to decrease meat consumption and the hypothetical reasons for doing so (iii, iv and v in Section 5).

The questionnaire was made available in Google Forms from the beginning of October of 2022 to mid-November and its link shared through the institutional emails from the pre-service teachers of both institutions. Two weeks after the first email, a reminder was sent and after four weeks, access to the questionnaire closed. The questionnaire, due to its division of the sections, allowed the participants to access the next section only after completing the previous one.

In the majority of the closed questions, the participants had to select the type of diet they followed, as well as the one they considered the best and the worst in the three dimensions of the study: the environment, animal welfare and human health. In the case of

the best or worst types of meat for human health, the participants had to choose between processed meat, read meat and poultry, and all have a similar impact (positive or negative). Finally, a Likert scale was used in the questions about the importance of meat for human health in general and for children in particular, and about the readiness of participants to reduce meat consumption. The score, with five, terms from total disagreement to total agreement, was applied to three statements related to these issues.

### 2.3. Validity and Reliability

The questionnaire was validated by two experts in Didactics of Science. A form was used where the experts inserted their opinion about the degree of relevance of each question, based on the aims of the study. The recommendations they made were related to the way certain questions were formulated and not to their content. One of the experts also suggested the inclusion in the questions about the types of meat considered the best and the worst for human health, with the option "all have the same positive or negative impact", which was accepted. A version was piloted by the research team on 12 volunteers from both institutions, not included in the final sample. No comprehension difficulties were detected, and this pilot administration also helped estimate the time needed by participants to answer the questionnaire, namely around 25 min. The average time information was included in the heading of the final questionnaire sent by email.

### 2.4. Data Analysis

As the number of participants per institution was unbalanced, the absolute frequencies of the results were converted in most cases into relative frequencies (percentages). For a few questions, to better compare the results between groups, inferential statistics were used. To verify the homogeneity between the groups concerning which types of meats were considered more and less healthy (1—processed meat, 2—red meat, 3—poultry, 4—no difference between them), a chi-square test was used. The four items were considered a nominal variable, since no hierarchy was established between them. To check the differences in the opinion of the participants about the importance of including meat in a diet (for adults and also for children and adolescents) and whether they considered they ate the amount of meat necessary for their wellbeing, a Mann–Whitney U test was used. In this case, the opinion about the three statements "Meat is essential for human health", "Meat is essential for the healthy growth of children" and "I eat the amount of meat necessary for my well-being" was given on a Likert scale from 1 (Strongly Disagree) to 5 (Totally agree) (ordinal variable). For all inferential statistics tests, the level of significance was 0.05 and the IBM SPSS Statistics 26 was used for these calculations.

The questions related to the best and worst diet for health, the environment and animal welfare, and the best and worst types of meat for health required a justification (open questions). The justifications were analysed through content analysis and similar ideas were counted as a single argument, even when expressed in a different way, e.g., "It is good for our health" or "it helps the body to function correctly". Even knowing that part of the richness of the arguments was lost, the grouping of similar ideas becomes essential for a more concise presentation of the results and a more focused approach to the main ideas presented instead of on the way they were presented. Certain answers were counted as redundant, and others not focused. In the first situation, there were answers that only repeated the question without advancing any explanation, e.g., "It is the best for the environment" or "It is processed meat because it is processed"; in the second case there were answers considered irrelevant to the formulated question, e.g., "It affects animal welfare" when the question asked about the worst diet for the environment.

The justifications were analyzed and grouped by each member of the research team separately and then compared. The interrater agreement Fleiss Kappa coefficient [46] was considered very good (0.93). One of the external experts was consulted to obtain a final consensus concerning a few disagreements. This consultation also took place in relation to the best statement to be adopted for similar ideas when expressed differently.

*2.5. Ethical Principles*

The questionnaire administered to pre-service teachers started with an explanation of the study in a concise way, so as to avoid giving too many clues about its content. It also included information from the research team and a guarantee of the total anonymity of the responses and none of the respondents' emails were associated with the responses of each questionnaire. The participants could also not complete the questionnaire before its submission and no registration of incomplete participations was implemented. The first question was an isolated section of the questionnaire to determine their informed consent. Only after this consent were the other sections of the questionnaire presented, one by one. The study received a unanimous favorable ethical opinion of the UTAD Ethics Committee (Doc30-CE-UTAD-2023).

It is important to state that in Portugal, pre-service teacher-training courses include the design and implementation of an educational study by each student, a condition for the conclusion of their master's degree. This requirement implies that the students are conscious of the importance of educational research. Therefore, they tend to accept participation in studies promoted by their own and other institutions.

**3. Results**

The diet options of the participants by institution are included in Table 1.

**Table 1.** The absolute and relative frequencies (%) of the different diets among the participants of both groups (UC—Urban Context; RC—Rural Context).

| Most Frequent Diet of the Participants by Institution | UC *n* = 129 | RC *n* = 68 |
|---|---|---|
| Lacto-ovo vegetarian | 6 (4.7) | 4 (5.9) |
| Omnivore with no restrictions | 100 (77.5) | 59 (86.8) |
| Vegan | 2 (1.6) | 0 |
| Lacto-ovo vegetarian and seafood | 7 (5.4) | 1 (1.4) |
| Omnivore with red meat restriction | 14 (10.8) | 4 (5.9) |

The most common diet mentioned by the participants is the omnivore without restrictions at a very high percentage, even higher in the RC (86.8% against 77.5 in the UC). On the contrary, a vegan diet was absent in the RC and residual in the UC. Only the omnivore with red meat restriction diet had some expression, particularly in the UC (10.8%).

Concerning the positive and negative impacts of the different diets on human health, the results follow a different trend from the participants' own diets (Tables 2 and 3).

**Table 2.** Opinion of the participants of both institutions (UC—Urban Context and RC—Rural Context) of the best diet for human health. The absolute and relative frequencies (%) of each diet and of the arguments to justify each selection are presented by institution (*n* = participants by institution).

| Best Diet for Human Health | UC *n* = 129 | RC *n* = 68 |
|---|---|---|
| **Omnivore with no restrictions** | **41 (31.7)** | **23 (33.8)** |
| - It is healthier, balanced and varied | 31 (24.0) | 13 (19.1) |
| - Redundant or undirected answer | 4 (3.1) | 1 (1.4) |
| - Did not know or did not justify | 6 (4.6) | 9 (13.2) |
| **Omnivore with red meat restriction** | **32 (25.0)** | **9 (13.2)** |
| - It is healthier, balanced and varied | 12 (9.4) | 5 (7.4) |
| - It is not necessary to eat meat | 1 (0,8) | 0 |
| - Redundant or undirected answer | 13 (10.2) | 3 (4.4) |
| - Did not know or did justify | 6 (4.6) | 1 (1.4) |
| **Lacto-ovo vegetarian** | **15 (11.6)** | **15 (22.1)** |
| - It is healthier, natural, balanced and varied | 6 (4.6) | 6(8.8) |
| - Meat and fish have microplastics | 1 (0.8) | 0 |
| - Redundant or undirected answer | 6 (4.6) | 5 (7.4) |

**Table 2.** *Cont.*

| Best Diet for Human Health | UC *n* = 129 | RC *n* = 68 |
|---|---|---|
| - Did not know or did not justify | 2 (1.6) | 4 (5.9) |
| **Lacto-ovo vegetarian and seafood** | **18 (14.0)** | **5 (7.4)** |
| - It is healthier, balanced and varied | 15 (11.6) | 3 (4.4) |
| - Redundant or undirected answer | 2 (1.6) | 0 |
| - Did not know or did not justify | 1 (0.8) | 2 (2.9) |
| **Vegan** | **23 (17.7)** | **16 (23.5)** |
| - It is healthier, balanced and varied | 11 (8.5) | 6 (8.8) |
| - It does not include animal food | 2 (1.6) | 4 (5.9) |
| - It is less contaminated (less chemicals and antibiotics) | 4 (3.1) | 0 |
| - It is without processed food | 2 (1.6) | 0 |
| - Redundant or undirected answer | 1 (0.8) | 2 (2.9 |
| - Did not know or did not justify | 3 (2.3) | 4 (5.9) |

Note: Percentages are shown to the first decimal place. Consequently, the sum of the percentages of the arguments sometimes differs slightly from the percentage of the diet to which they are related.

**Table 3.** Opinion of the participants of both institutions (UC—Urban Context and RC—Rural Context) on the worst diet for human health. The absolute and relative frequencies (%) of each diet and the arguments to justify each selection are presented by institution (*n* = participants by institution).

| Worst Diet for Human Heath | UC *n* = 129 | RC *n* = 68 |
|---|---|---|
| **Omnivore with no restrictions** | **69 (53.5)** | **35 (51.5)** |
| - It includes red and processed meat | 30 (23.1) | 12 (17.7) |
| - Animals can transmit diseases | 1 (0.8) | 0 |
| - It causes cardiovascular diseases and obesity | 2 (1.6) | 0 |
| - It has no restrictions (more unhealthy food) | 6 (4.6) | 9 (13.2) |
| - It is more contaminated (more chemicals and antibiotics) | 6 (4.6) | 1 (1.4) |
| - Redundant or undirected answer | 6 (4.6) | 4 (5.9) |
| - Did not know or did not justify | 18 (14.1) | 9 (13.2) |
| **Omnivore with red meat restriction** | **9 (7.0)** | **2 (3.0)** |
| - If in excess red meat causes diseases | 2 (1.6) | 1 (1.4) |
| - Redundant or undirected answer | 1 (0.8) | 0 |
| - Did not know or did not justify | 6 (4.6) | 1 (1.4) |
| **Lacto-ovo vegetarian** | **6 (4.7)** | **4 (5.8)** |
| - It lacks necessary nutrients | 5 (3.9) | 1 (1.4) |
| - Il includes a lot of certain foods like cheese | 1 (0.8) | 0 |
| - Redundant or undirected answer | 0 | 0 |
| - Did not know or did not justify | 0 | 3 (4.4) |
| **Lacto-ovo vegetarian and seafood** | **4 (3.1)** | **2 (3.0)** |
| - It lacks necessary nutrients | 2 (1.6) | 0 |
| - Redundant or undirected answer | 0 | 1 (1.4) |
| - Did not know or did not justify | 2 (1.6) | 1 (1.4) |
| **Vegan** | **41 (31.7)** | **25 (36.7)** |
| - It excludes certain nutrients (animal proteins, vitamins, iron) | 24 (18.5) | 19 (27.9) |
| - It forces people to take food supplements | 5 (3.9) | 2 (2.9) |
| - Redundant or undirected answer | 2 (1.6) | 0 |
| - Did not know or did not justify | 10 (7.7) | 4 (5.9) |

Note: Percentages are shown to the first decimal place. Consequently, the sum of the percentages of the arguments sometimes differs slightly from the percentage of the diet to which they are related.

The omnivore diet with no restrictions was considered the best for human health by both groups, with very close percentages (31.7% in the UC and 33.8% in the RC). The main argument was that it is healthier, balanced and varied. Even so, these percentages corresponded to less than half of those who said they followed this diet daily, as shown by the results included in Table 1. The opposite case was found for the vegan diet, practically absent as a daily diet of the participants, but considered healthy by 17.7 of the UC and 23.5 of the RC. The worst diet selected by the participants among the five proposed was the omnivore with no restrictions (53.5%, in the UC and RC, 51.5%), mainly because it includes

unhealthy types of meat (red and processed) and favors the ingestion of more unhealthy foods, since it does not have restrictions. Even so, the vegan diet was not considered particularly healthy by many participants of both groups (31.7 in the UC and 36.7 in the RC). The main reason was the lack of certain nutrients, which are absent in plants or are more difficult to obtain.

When concerning the positive and negative impacts of the proposed diets on the environment, the participants seem very critical about their favorite diet (Tables 4 and 5).

**Table 4.** Opinion of the participants of both institutions (UC—Urban Context and RC—Rural Context) of the best diet for the environment. The absolute and relative frequencies (%) of each diet and the arguments to justify each selection are presented by institution (*n* = participants by institution).

| Best Diet for the Environment | UC *n* = 129 | RC *n* = 68 |
|---|---|---|
| **Omnivore with no restrictions** | **5 (3.9)** | **5 (7.4)** |
| - It is more balanced | 0 | 2 (2.9) |
| - Redundant or undirected answer | 5 (3.9) | 1 (1.4) |
| - Did not know or did not justify | 0 | 2 (2.9) |
| **Omnivore with red meat restriction** | **6 (4.7)** | **0** |
| - It decreases production to feed animals | 2 (1.6) | 0 |
| - It reduces meat consumption | 1 (0.8) | 0 |
| - It reduces pollution and greenhouse gases | 2 (1.6) | 0 |
| - Redundant or undirected answer | 1 (0.8) | 0 |
| **Lacto-ovo vegetarian** | **19 (14.7)** | **14 (20.5)** |
| - It decreases pollution | 2 (1.6) | 3 (4.4) |
| - It decreases the impact of industries related to meat | 7 (5.0) | 4 (5.9) |
| - Redundant or undirected answer | 7 (5.0) | 3(4.4) |
| - Did not know or did not justify | 3 (2.3) | 4 (5.9) |
| **Lacto-ovo vegetarian and seafood** | **7 (5.4)** | **3 (4.4)** |
| - It decreases the production and consumption of meat | 3 (2.3) | 2 (2.9) |
| - Redundant or undirected answer | 2 (1.6) | 1 (1.4) |
| - Did not know or did not justify | 2 (1.6) | 0 |
| **Vegan** | **92 (71.3)** | **46 (67.7)** |
| - No meat consumption | 33 (25.6) | 10 (14.7) |
| - It decreases pollution | 19 (15.0) | 11 (16.2) |
| - It reduces resource depletion | 4 (3.1) | 1 (1.4) |
| - It reduces costs and water | 5 (3.9) | 0 |
| - It reduces mass-production | 0 | 2 (2.9) |
| - Redundant or undirected answer | 21 (16.0) | 9 (13.2) |
| - Did not know or did not justify | 10 (7.7) | 13 (19.1) |

Note: Percentages are shown to the first decimal place. Consequently, the sum of the percentages of the arguments sometimes differs slightly from the percentage of the diet to which they are related.

**Table 5.** Opinion of the participants of both institutions (UC—Urban Context and RC—Rural Context) of the worst diet for the environment. The absolute and relative frequencies (%) of each diet and the arguments to justify each selection are presented by institution (*n* = participants by institution).

| Worst Diet for the Environment | UC *n* = 129 | RC *n* = 68 |
|---|---|---|
| **Omnivore with no restrictions** | **113 (87.6)** | **52 (76.4)** |
| - It increases animal production and meat consumption | 12 (9.3) | 5 (7.4) |
| - It causes more pollution through more polluting processes | 64 (49.6) | 22 (32.3) |
| - It increases resource depletion | 5 (3.9) | 2 (2.9) |
| - It increases water consumption | 2 (1.6) | 0 |
| - Redundant or undirected answer | 22 (17.0) | 11 (16.2) |
| - Did not know or did not justify | 8 (6.2) | 12 (17.7) |
| **Omnivore with red meat restriction** | **4 (3.1)** | **4 (5.9)** |
| - It causes more pollution and greenhouse gases | 4 (3.1) | 2(2.9) |
| - Redundant or undirected answer | 0 | 2 (2.9) |
| - Did not know or did not justify | 0 | 0 |

**Table 5.** *Cont.*

| Worst Diet for the Environment | UC *n* = 129 | RC *n* = 68 |
|---|---|---|
| **Lacto-ovo vegetarian** | **5 (3.9)** | **3 (4.4)** |
| - It causes more species extinction | 1 (0.8) | 0 |
| - It increases excessively the consumption of vegetables | 2 (1.6) | 0 |
| - Redundant or undirected answer | 1 (0.8) | 1 (1.4) |
| - Did not know or did not justify | 1 (0.8) | 2 (2.9) |
| **Lacto-ovo vegetarian and seafood** | **3 (2.3)** | **5 (7.4)** |
| - It causes greater extinction of marine species | 2 (1.6) | 1 (1.4) |
| - It increases pollution caused by fishing boats | 0 | 1 (1.4) |
| - Redundant or undirected answer | 1 (0.8) | 0 |
| - Did not know or did not justify | 0 | 3 (4.4) |
| **Vegan** | **4 (3.1)** | **4 (5.9)** |
| - It decreases animal production and causes less pollution | 0 | 2 (2.9) |
| - Redundant or undirected answer | 4 (3.1) | 1 (1.4) |
| - Did not know or did not justify | 0 | 1 (1.4) |

Note: Percentages are shown to the first decimal place. Consequently, the sum of the percentages the arguments sometimes differs slightly from the percentage of the diet to which they are related.

Only 3.9% of the participants of the UC and 7.4% of the RC considered the omnivore diet with no restrictions the best diet for the environment. Consequently, 87.6% and 76.4%, respectively, considered it the worst. The main reasons were the industrial production processes that cause more pollution and their association with the mutually influencing cycle between more production and consumption. On the contrary, the vegan diet was considered the best for the environment by 71.3% of the UC and 67.7% of the RC. The reasons are in agreement with the criticisms for the omnivorous diet, since the vegan diet excludes meat and implies less pollution. The lacto-ovo vegetarian diet was selected as the second best one for the environment by 14.7% of the UC and 20.5% of the RC, based on similar arguments to the ones used for the vegan diet.

Finally, the opinion about the best and worst diets for animal welfare is systematized in Tables 6 and 7.

**Table 6.** Opinion of the participants of both institutions (UC—Urban Context and RC—Rural Context) of the best diet for animal welfare. The absolute and relative frequencies (%) of each diet and the arguments to justify each selection are presented by institution (*n* = participants by institution).

| Best Diet for Animal Welfare | UC *n* = 129 | RC = 68 |
|---|---|---|
| **Omnivore with no restrictions** | **4 (3.1)** | **2 (3.0)** |
| - Animals are not badly treated | 1 (0.8) | 0 |
| - Redundant or undirected answer | 3 (2.3) | 2 (2.9) |
| **Omnivore with red meat restriction** | **2 (1.6)** | **0** |
| - Did not know or did not justify | 2 (1.6) | 0 |
| **Lacto-ovo vegetarian** | **14 (10.8)** | **13 (19.1)** |
| - It does not harm animals | 4 (3.1) | 0 |
| - It does not involve direct consumption of animals | 2 (1.6) | 2 (2.9) |
| - It needs less animal production | 1 (0.8) | 1 (1.4) |
| - Redundant or undirected answer | 1 (0.8) | 1 (1.4) |
| - Did not know or did not justify | 6 (4.6) | 9 (13.2) |
| **Lacto-ovo vegetarian and seafood** | **1 (0.8)** | **3 (4.4)** |
| - Redundant or undirected answer | 0 | 2 (2.9) |
| - Did not know or did not justify | 1 (0.8) | 1 (1.4) |
| **Vegan** | **108 (83.7)** | **50 (73.5)** |
| - It respects animals since they are not consumed | 84 (65.1) | 36 (52.9) |
| - It does not affect animals directly, but soy plantations destroy animal habitats | 1 (0.8) | 0 |
| - Redundant or undirected answer | 8 (6.2) | 8 (11.8) |
| - Did not know or did not justify | 15 (11.6) | 16 (23.6) |

Note: Percentages are shown to the first decimal place. Consequently, the sum of the percentages of the arguments sometimes differs slightly from the percentage of the diet to which they are related.

**Table 7.** Opinion of the participants of both institutions (UC—Urban Context and RC—Rural Context) of the worst diet for animal welfare. The absolute and relative frequencies (%) of each diet and the arguments to justify each selection are presented by institution (*n* = participants by institution).

| Worst Diet for Animal Welfare | UC *n* = 129 | RC *n* = 68 |
|---|---|---|
| **Omnivore with no restrictions** | **118 (91.4)** | **60 (88.1)** |
| - It increases animal deaths | 45 (34.8) | 21 (30.9) |
| - It involves bad production conditions (lack of space, bad treatment, disrespect for life cycle of animals) | 33 (25.5) | 18 (26.5) |
| - It uses the animals instrumentally | 2 (1.6) | 2 (2.9) |
| - It increases the extinction of species | 2 (1.6) | 2 (2.9) |
| - It increases the number of animals consumed | 1 (0.8) | 0 |
| - Redundant or undirected answer | 9 (7.0) | 7 (10.3) |
| - Did not know or did not justify | 26 (20.2) | 10 (14.7) |
| **Omnivore with red meat restriction** | **7 (5.4)** | **1 (1.5)** |
| - It provokes bad conditions of production | 1 (0.8) | 0 |
| - It leads to the death of many animals | 1 (0.8) | 0 |
| - It consumes many animals | 2 (1.6) | 0 |
| - Redundant or undirected answer | 3 (2.3) | 1 (1.4) |
| **Lacto-ovo vegetarian** | **1 (0.8)** | **2 (3.0)** |
| - It provokes the death of many animals | 0 | 1 (1.4) |
| - Redundant or undirected answer | 0 | 1 (1.4) |
| - Did not know or did not justify | 1 (0.8) | 0 |
| **Lacto-ovo vegetarian and seafood** | **1 (0.8)** | **3 (4.4)** |
| -It increases the extinction of fish species | 1 (0.8) | 1 (1.4) |
| -It promotes the keeping of animals in restaurant aquariums | 0 | 1 (1.4) |
| - Did not know or did not justify | 0 | 1 (1.4) |
| **Vegan** | **2 (1.6)** | **2 (3.0)** |
| - Redundant or undirected answer | 0 | 0 |
| - Did not know or did not justify | 2 (1.6) | 2 (2.9) |

Note: Percentages are shown to the first decimal place. Consequently, the sum of the percentages of the arguments sometimes differs slightly from the percentage of the diet to which they are related.

The vegan diet was considered the best for animal welfare by 83.7% of the participants of UC and 73.5% of the RC, respectively, since it excludes animal consumption and, consequently, respects the integrity of animals. Even so, one participant mentioned that soy plantations can be responsible for the destruction of animal habitats, which is an interesting idea, but without any expression in the global sample. Unsurprisingly, the omnivore diet with no restrictions was considered the worst by 91.4% of the participants of the UC and 88.1% of the RC. The mains reasons were related to the number of deaths it provokes, as well as the fact that poor production processes have no respect for the fundamental needs of animals.

It is also relevant to mention that several participants selected the impact of the different diets on human health, the environment and animal welfare without giving a justification or by giving a redundant or undirected answer. This may mean an absence of clear ideas to support their selections.

Considering the best and worst types of meat for human health, the results were very similar in both groups (Tables 8 and 9).

**Table 8.** Opinion of the participants of both institutions (UC—Urban Context and RC—Rural Context) of the type of meat better for health. The absolute and relative frequencies (%) of each type of meat and the arguments to justify each selection are presented by institution (*n* = participants by institution). The arguments to justify each type correspond only to the absolute frequencies for both institutions.

| The Type of Meat Better for Health | UC *n* = 129 | RC *n* = 68 |
|---|---|---|
| **Processed meat** | **2 (1.6)** | **0** |
| - The diet must be varied | 1 (0.8) | 0 |
| - Did not know or did not justify | 1 (0.8) | 0 |

**Table 8.** *Cont.*

| The Type of Meat Better for Health | UC *n* = 129 | RC *n* = 68 |
|---|---|---|
| **Red meat** | **2 (1.6)** | **5 (7.4)** |
| -It has less chemicals/hormones | 1 (0.8) | 1 (1.4) |
| -It has different nutrients | 0 | 1 (1.4) |
| - Did not know or did not justify | 1 (0.8) | 3 (4.4) |
| **Poultry** | **119 (92.1)** | **62 (91.1)** |
| - It has less fat | 32 (25.0) | 28 (41.2) |
| - It is healthier | 26 (20.1) | 10 (14.7) |
| - It poses less risks | 3 (2.3) | 0 |
| - It is more nutritive and balanced | 6 (4.6) | 3 (4.4) |
| - It is more digestible | 2 (1.6) | 0 |
| - It is less processed | 3 (2.3) | 5 (7.4) |
| -It involves less chemicals/hormones | 2 (1.6) | 0 |
| -It doesn't cause cardiovascular diseases | 1 (0.8) | 0 |
| - Redundant or undirected answer | 14 (10.7) | 4 (5.9) |
| - Did not know or did not justify | 30 (23.2) | 12 (17.7) |
| **All have a positive impact** | **6 (4.7)** | **1 (1.5)** |
| - All have important properties | 4 (3.1) | 0 |
| - Did not know or did not justify | 2 (1.6) | 1 (1.4) |

Note: Percentages are shown to the first decimal place. Consequently, the sum of the percentages of the arguments sometimes differs slightly from the percentage of the type of meat to which they are related.

**Table 9.** Opinion of the participants of both institutions (UC—Urban Context and RC—Rural Context) of the type of meat is worst for health. The absolute and relative frequencies (%) of each type of meat and the arguments to justify each selection are presented by institution (*n* = participants by institution).

| The Type of Meat Worst for Health | UC *n* = 129 | RC *n* = 68 |
|---|---|---|
| **Processed meat** | **103 (79.7)** | **56 (82.4)** |
| -Due to processes of production and conservation (additives, preservatives such as salt, etc.) | 33 (25.5) | 22 (32.3) |
| - It includes the less noble parts of animals | 3 (2.3) | 4 (5.9) |
| - It has unknown constituents | 4 (3.1) | 2 (2.9) |
| - It mixes constituents | 5 (3.9) | 1 (1.4) |
| - It has more fat | 9 (7.0) | 4 (5.9) |
| - It is less healthy | 6 (4.6) | 2 (2.9) |
| - It is not natural | 5 (3.9) | 1 (1.4) |
| - It provokes diseases (cancer, cardiovascular) and food poising | 3 (2.3) | 4 (5.9) |
| - Redundant or undirected answer | 18 (14.0) | 6 (8.8) |
| - Did not know or did not justify | 17 (13.1) | 10 (14.7) |
| **Red meat** | **21 (16.3)** | **12 (17.6)** |
| - It has more fat | 3 (2.3) | 4 (5.9) |
| - It is unhealthier | 4 (3.1) | 1 (1.4) |
| - It makes digestion more difficult | 0 | 2 (2.9) |
| - It provokes diseases (cancer, obesity) | 3 (2.3) | 0 |
| - Redundant or undirected answer | 4 (3.1) | 2 (2.9) |
| - Did not know or did not justify | 7 (5.4) | 3 (4.4) |
| **Poultry** | **0** | **0** |
| **All have a negative impact** | **5 (3.9)** | **0 (0)** |
| - All use chemicals | 1 (0.8) | 0 |
| -When consumed excessively | 1 (0.8) | 0 |
| - Redundant or undirected answer | 1 (0.8) | 0 |
| - Did not know or did not justify | 2 (2.3) | 0 |

Note: Percentages are shown to the first decimal place. Consequently, the sum of the percentages of the arguments sometimes differs slightly from the percentage of the type of meat to which they are related.

Consequently, no statistical differences were detected after the application of a chi-square for the worst types of meat ($p$ = 0.257) and the conditions of applicability of this test for the best types of meat were not met, since the minimum expected count was less than 1.

Poultry was considered the best meat for human health, with a percentage higher than 90% in both groups. It was considered healthier, with less fat, and more nutritive and balanced. Processed meat was considered the worst type in both groups, with percentages around 80%. The main argument was centered on the processes of production and conservation, which include many chemicals, additives and preservatives. At a much lower frequency, but also interesting, was the argument that this kind of meat mixes a lot of constituents, some of them unknown, in the assumption that this can be related to the health problems it causes. Concrete diseases, such as cancer, cardiovascular diseases and food poisoning, were only mentioned by a few participants.

As in the previous questions, a considerable number of participants revealed difficulty in justifying their choices.

Finally, Table 10 evaluates to what extent the participants of each institution agree with the following three statements: meat is important for adults; meat is important for children; I consume the amount of meat necessary for my wellbeing.

**Table 10.** The degree of agreement of the participants of both institutions (UC—Urban Context and RC—Rural Context) on a Likert scale (1—Totally disagree to 5—Totally agree) about the three statements related to the importance of meat for adults, children and in personal terms. The median by institution is also included, as well as the $p$-value after the application of a Mann–Whitney U test to check statistically significant differences between the groups.

| | | UC | | | | | RS | | | UC | RS | |
|---|---|---|---|---|---|---|---|---|---|---|---|---|
| 1-TD | 2 | 3 | 4 | 5-TA | 1-TD | 2 | 3 | 4 | 5-TA | Median | | $p$ |
| | | | | | **Meat is important for adults** | | | | | | | |
| 21 | 14 | 47 | 32 | 15 | 1 | 9 | 25 | 26 | 7 | 3 | 3 | 0.04 |
| (16.3) | (10.9) | (36.4) | (24.8) | (11.6) | (1.5) | (13.2) | (36.8) | (38.2) | (10.3) | | | |
| | | | | | **Meat is important for children** | | | | | | | |
| 12 | 18 | 40 | 37 | 22 | 2 | 5 | 23 | 24 | 14 | 3 | 4 | 0.73 |
| (9.3) | (14.0) | (31.0) | (28.7) | (17.1) | (2.9) | (7.4) | (33.8) | (35.3) | (20.6) | | | |
| | | | | | **I eat the necessary amount of meat for my well-being** | | | | | | | |
| 11 | 12 | 30 | 45 | 31 | 0 | 4 | 27 | 20 | 17 | 4 | 4 | 0.68 |
| (8.5) | (9.3) | (23.3) | (34.9) | (24.0) | 0 | (5.9) | (39.7) | (29.4) | (25.0) | | | |

Both groups considered that meat is important for adults. Even so, the participants of the UC agreed less with this idea and the differences between the groups were statistically different ($p$ = 0.04). In the case of the importance of meat for children's health, there were no statistical differences between the two groups (0.73). Even so, the participants from the RC tend to agree more with this importance, which is expressed by a higher median (4 instead of 3). The participants from both groups tend to agree that they eat the amount of meat necessary for their well-being. Nevertheless, it is impossible to infer whether the participants actually eat a lot of meat. That is why the next question tried to find out if they would be willing to reduce meat consumption and, in that case, what would be the reasons for doing so. In this case, they could select two reasons among seven, including the possibility of considering no need to reduce meat consumption. These results are presented in Table 11.

Almost 80% of the participants from both institutions considered the possibility of reducing meat consumption. Consequently, nearly 20% of the pre-service teachers from both groups considered that there was no need to reduce the meat in their diet. Willingness to do this would be determined by the negative impact of meat on the environment and animal welfare, both of which were rated within a range of 30% to 40%. The impact of meat on human health was a reason cited far less frequently by the participants from both

groups and the other reasons were even less frequent, with very low percentages when considering their first or second option.

**Table 11.** Predisposition of the participants of both institutions (UC—Urban Context and RC—Rural Context) to reduce meat consumption. The absolute and relative frequencies (%) of the reasons given as the first and second option.

| Reasons for Meat Reduction | First Option | | Second Option | |
| --- | --- | --- | --- | --- |
| | UC | RC | UC | RC |
| Health | 16 (12.4) | 8 (11.8) | 11 (8.5) | 4 (5.9) |
| Price | 0 | 2 (2.9) | 1 (0.8) | 2 (2.9) |
| Animal welfare | 40 (31.0) | 18 (26.5) | 42 (32.6) | 25 (36.8) |
| Environment | 42 (32.6) | 26 (38.2) | 44 (34.1) | 19 (27.9) |
| Religious motives | 1 (0.8) | 0 | 1 (0.8) | 1 (1.5) |
| Unsafe food | 1 (0.8) | 0 | 1 (0.8) | 3 (4.4) |
| No need to reduce | 29 (22.4) | 14 (20.6) | 29 (22.4) | 14 (20.6) |

## 4. Discussion

The participants of both groups, UC and RC, follow an omnivorous diet with no restrictions, an option selected by the great majority of them, even more pronounced in the rural group. If it is true that in an urban context people normally have more food options, it is also true that food distribution in more recent decades has permitted access to more different types of food, even in smaller cities. Thus, social–cultural factors can certainly better explain this difference.

Consequently, the other diets were rarely selected, including plant-based diets (lacto-ovo, vegetarian and vegan). A study promoted by the Centro Vegetariano (Vegetarian Center) in 2017 [47], with a random and representative sample, concluded that 1.2% of the Portuguese are vegetarians and 0.6% vegans, a lower percentage when compared with the vegetarians in other European countries (e.g., 9% in Germany and 3% in the United Kingdom) [48]. However, if the Portuguese percentages appear to be low, it should be noted that compared to another study implemented in 2007 by the same entity, the number of vegetarians quadrupled and vegans doubled. Thus, the results of the present study, although not using a representative sample of the country, are somehow in line with the results of the 2017 study. Another study in an urban Portuguese higher education institution also found similar results [49].

The respondents from both groups, when asked about the healthiest diet, continue to select the omnivore with no restrictions, but with a far smaller percentage when compared to those who claimed to follow this diet. This means that they consider that all other diets are somehow healthier, since the choice of the omnivorous diet with no restrictions was around 32% of the answers obtained.

An even more divergent result from their own food options was the consideration that the omnivorous with no restrictions diet has the greatest negative impact on the environment and animal welfare. The reasons stated by the participants for considering this diet the worst for the environment were not particularly detailed scientifically. The participants tended to mention the increase in pollution in general and only a few made a connection to climate change, although omitting the concrete designation of greenhouse gases. A study with pre-service teachers from Portugal and Spain, mainly focused on identifying their knowledge of the impact of livestock production on global warming, also concluded that students have difficulties in referring to concrete greenhouse gases [50]. Besides this result, the majority of the respondents admitted to following a diet they do not consider particularly healthy and which proves to be even more harmful in the other two dimensions under discussion. This contradiction is part of the so-called meat paradox, a term that designates the dissonance between the frequent consumption of meat and the recognition that it has a number of harmful effects [51].

Joy [52] uses the term carnism applied to meat consumption in opposition to vegetarianism, an ideology where eating animals is considered an ethical choice, based on three justifications: it is normal, a social norm that describe how the majority of people behave; it is natural, since meat is part of an omnivorous diet selected during human evolution; it is necessary, as a guarantee of human health. Many other reasons support meat consumption. For instance, right-wing political options tend to favor animal exploitation and high meat consumption [53].

The pre-service teachers surveyed from both groups showed a predisposition to reduce meat consumption for reasons centered on the environment and animal welfare. This result is partly in line with the one obtained in a study carried out in Denmark, with a sample of 85% of females, that concluded that animal production processes and animal welfare were the main reasons for the decrease in meat consumption [28], and may result from the similarities of both sample studies, mainly with females.

In the Portuguese context, those that are most informed about environmental issues are more aware of the impacts of different diets and tend to choose a diet more based on legumes and vegetables, and less on poultry and livestock, while without completely eliminating meat consumption [54]. A study in Germany also found similar results [55]. Even so, the contradictions found in the present study just show how difficult it can be to reduce meat consumption even when the need arises. In fact, social and cultural factors make food choices mostly unconscious, since they are acquired from family and social circles and are based on habitats, tradition and past behavior [56].

The participants of the present study also assessed different types of meat according to the recommendations of The School of Public Health of Harvard [57], considering poultry healthier than red and processed meat and supporting their choices with relevant arguments. To decrease processed, but also red meat, it seems particularly important to know the dangers associated with its consumption, which was only in part revealed by the participants. Even so, the dangers associated with poultry consumption were completely absent, the participants highlighting only the positive features associated with this type of meat. In the case of processed meat, it is important to remember that there is a historical tradition associated with different processes for keeping food consumable for a long period. Although this tradition continues to exist, many foods included in this category are now produced to seduce the consumer based on the taste for a growing range of products [13].

In this study, the participants also considered meat consumption especially important for children, with a higher agreement among those from the rural context. A similar result was found by Cmic [58] in Slovenia, with a representative sample of adults from the two major cities in the country. He concluded that the respondents considered a vegetarian diet healthy for adults, but not for children.

The recommendations of The School of Public Health of Harvard [57] propose that young people eat more vegetables without totally eliminating the consumption of animal products. They also, however, recommend less red meat (beef, pork, lamb), and the avoidance of processed meats (bacon, deli meats, hot dogs, sausages). Similar guidance can be found from other authors who consider an omnivorous diet advisable in childhood and adolescence or, at least, a lacto-ovo vegetarian diet, even recognizing that a vegetarian or vegan diet could be safe if carefully designed. In fact, animal protein seems beneficial for young people and for those subjected to physical demands due to its effects on body composition and muscle strength, and any plant-based diet should find a supply for the nutrients exclusively present in meat and in which amount they are particularly beneficial to the body [59,60]. Even so, the results of meat consumption in children and adolescents in Western countries are mostly in excess of double or even triple the recommended quantity [61], a main reason for the presence of this subject in the primary school curriculum, even knowing that children's families are central in shaping their attitudes toward certain foods and even diets; parents with a higher consumption of meat, for instance, influence the taste and habitats of their children [28].

Globally, given the answers to the questionnaire and especially based on the justifications presented, it was possible to verify that many participants have an idea about the impact of the different diets on human health, the environment and animal welfare, as well as some knowledge regarding the best and worst types of meat for human health. Even so, the number of respondents who did not justify their ideas or gave redundant and undirected answers was also high. Some justifications were not relevant and revealed wrong ideas, although in this case to a lower extent.

There is, therefore, an undeniable need to increase pre-service teachers' knowledge about the subject under discussion, highlighting different ways to achieve this aim and also contribute to a shift in dietary habits, a change considered essential to mitigate global warming and support the increase in population by minimizing its impact on the planet [62].

Thus, in order to reduce meat consumption, it seems essential to act on the role that unconscious and conscious choices play in choosing food. In the first case, given that the choices are immediate and made without thought, it is important to act on marketing and advertising strategies and also on the way in which plant and animal foods are presented in supermarkets, on restaurant menus and in canteens; in the second case, people move by values and intentions, and are influenced by education and information, which highlights the importance of formal education [15]. Even so, it is important not to forget that diet changes are normally slow, not easy, but also not impossible.

The most popular ways to reduce the consumption of red and processed meat is through food labelling and media campaigns, and by reinforcing consumer information. A study in Sweden showed the effectiveness of giving information about the impact of different kinds of food in an environment based on carbon footprint labels, but this success is mainly among those who are used to making more sustainable food choices [63]. On the other hand, targeting industrial actions, such as advertising bans, or imposing financial penalties, such as increasing prices, were much less popular. It was also found that the acceptability of the different policies did not differ when the focus of reducing meat consumption was justified by its positive effect on human health or on the environment [61]. These findings help reinforce the importance of higher education in the reduction of meat consumption, since formal education aims to prepare students to critically analyze information from different sources and is focused on giving reliable information and data, which seems closer to the policies more popular with consumers.

Higher education occurs frequently at the beginning of adulthood and is a sensitive period in a person's life. In this new experience, the students face all the constraints motivated by their academic tasks and timetables which tend to decrease the quality of their meals. They enter a new and sensitive period of their lives. However, as Koster [56] states, all the sensitive periods of life can provoke changes and re-orientations, including in food habitats, helping promote a better understanding of meat consumption in human health, the environment and animal welfare, and how these three dimensions can sometimes collide, imposing priorities and commitments. Even so, a coherent approach in higher education to the present issue will show why a reduction in meat consumption can be beneficial to human health, the environment and animal welfare. It should cover how the dangers of meat consumption can be minimized through the ingestion of different foods in a meal, which can also affect bioaccessibility and bioavailability of the nutrients. It should further discuss the relevance of different diets according to the age of the individuals, state of health, activity, sex and other variables, reasons why it is difficult to impose an exact daily or weekly recommendation concerning meat intake, even knowing that certain values appear in a number of publications. It is also relevant to discuss the role that animal production can have in certain economically deprived areas and to identify sustainable ways of rearing animals and different ways to guarantee their welfare. Just by way of an example, certain production measures can reduce the methane emissions of ruminants and include a better breed selection of the animals (even knowing that this selection is not always a guarantee of animal welfare), a better composition of the plants included in their diet or even the increase in productivity per animal [64]. Moreover, the red meat climate

footprint, based on greenhouse gas emissions, is higher when cattle are raised in extensive grazing herds instead of intensive factoring units. However, when considering animal welfare, the first husbandry system is far better [29].

Despite the importance given to the role of education in food choices within the three dimensions under discussion, it is important to point out that the relationship between knowledge and consistent action is complex and sometimes tenuous. However, knowledge facilitates change, especially in people who are aware of the problems and have a willingness to change [9]. In addition, it poses several didactic challenges about the best ways to approach the present issue in a formal context.

## 5. Limitations of the Study

Despite the fact that most respondents claim to follow an omnivorous diet with no restrictions, there were no questions about the amount of red and processed meat actually consumed (or of other types of food). This can be considered a limitation of the study, since the proportion of meat when compared with other foods in the daily diet of the participants is unknown. Even so, studies that have sought to question participants about the intake amount of each food found that the respondents normally have difficulty in answering and giving accurate information. This is mainly because they lack the right perception of the exact quantity of the intake of each food.

Additionally, some justifications for answers to open questions were not very detailed, although in surveys with the present features—several sections and a high number of questions—there is often a tendency for respondents to be concise, which prevents a complete and correct assessment of their knowledge.

The sample is mainly composed of females, a consequence of the low number of males attending pre-service teachers' courses in Portugal. In fact, the comparison, according to sex, of the perceptions and knowledge of meat consumption was not possible, but could be interesting, since meat is frequently associated with masculinity.

Another limitation stems from the comparison between the groups. In fact, one of the institutions is in the largest metropolitan area of the country and the other in a smaller city. This smaller city is a district capital, and although many students are from the surrounding rural areas, it is still a city with access to many of the commercial structures that normally exist in an urban environment and also offer restaurants from other cultures with different types of food. Thus, the comparison between urban and rural contexts is imperfect and could be a main reason for the great similarity between the results of the two groups. However, as there are no higher education institutions in totally rural areas, our choice is justified.

## 6. Conclusions

The present study verified that the most frequent diet among the respondents of both groups (urban and rural) is omnivorous with no restrictions, mentioned even more frequently by the second group. However, this diet was considered the worst for the environment and animal welfare, but the best for human health. In the latter case, the frequency of responses did not stand out from other diets considered. The respondents agree that meat consumption is important for human health, especially for children, a more significant result in the participants from the rural context. However, the latter were aware of some of the harmful effects of red and processed meat, and manifested a predisposition to reduce meat consumption in general based on environmental and animal welfare factors. Although a majority of respondents justified their choices, the answers were not particularly detailed. In addition, several students stated that they did not know how to justify their choices, did not answer at all or gave inconsistent answers. That is why it is considered that the topic deserves a greater emphasis during teacher training courses, especially because the topic of food is part of the curriculum in the years of schooling for which these trainees are being prepared.

**Author Contributions:** Conceptualization, A.A. and I.R.; methodology, A.A., I.R. and J.T.; software, A.A.; validation, A.A. and I.R.; formal analysis, A.A. and I.R.; investigation, A.A., I.R. and J.T.; resources, A.A. and I.R.; data curation, A.A., I.R. and J.T.; writing—original draft preparation, A.A.; writing—review and editing, A.A., I.R. and J.T.; visualization, A.A., I.R. and J.T.; supervision, A.A., I.R. and J.T.; project administration, A.A., I.R. and J.T.; funding acquisition, I.R. All authors have read and agreed to the published version of the manuscript.

**Funding:** This research received no external funding.

**Institutional Review Board Statement:** The study received ethical approval from the Ethics Commission of University of Trás-os-Montes e Alto Douro, file number Doc30-CE-UTAD-2023, and date of approval: 25 May 2023.

**Informed Consent Statement:** Informed consent was obtained from all subjects involved in the study.

**Data Availability Statement:** The data presented in this study are available on request from the corresponding author.

**Conflicts of Interest:** The authors declare no conflict of interest.

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
