# Peer review of "The Impact of Meat Consumption on Human Health, the Environment and Animal Welfare: Perceptions and Knowledge of Pre-Service Teachers"

_societies, doi:10.3390/soc13060143_

Round 1

Author Response

The changes and comments concerning the comments of the reviewer are the following:

  1. The reference has been substituted by another from a scientific article.

  1. A few verb tenses have been changed when considered relevant

  1. It is stated on the homepage of the journal that any coherent system of references is accepted. Even so, the references have been changed but we would suggest a clarification of the journal on its homepage about this issue.

  1. The authors quoted between 3 and 6 state the relationship between meat consumption and economic development, independently of the main focus of the article or book, the reason why they are cited. The article of Crepaldi Lunkes et al. is a study about the consumption of different types of meat food including fish, pork, beef and poultry. They compared this consumption according to age, level of schooling, economic level and sex. They conclude that red meat is highly consumed especially by the young and even by those in higher school. On the contrary, fish is much less consumed. Therefore, the comment of the reviewer about the article is not correct.

  1. Thanks for recognizing this aspect.

  1. The section 1.1 was excluded.

  1. You are right. It is an exploratory study and that information is now clearly stated.

  1. Primary school courses can have several designations in Portugal, since they can prepare only for the first cycle of primary school (four years) or for the first and second cycles of primary school (six years, being the last two disciplinary with various specialized teachers). Therefore, in our opinion, it is not important to state the exact names of the courses. Also for ethical reasons, it is better not to associate the respondents with any particular course.

  1. The type of questions of the questionnaire is now clearly stated. The fifth objective of the study is related to the reflection of the research team had done based on the results obtained for teacher training courses. Therefore, it was not an objective to be included in the questionnaire. The study was designed to check the knowledge of pre-service teachers about the issues approached, with the main intention of promoting changes in the syllabus of the curricular units of their courses. That is the reason why questions about meat consumption and food education in primary school were not included.

  1. The experts were from the field of science education, since environmental issues, animal welfare issues, human health and diet options are approached in science and environment departments in both institutions involved in the study. A few more details of the criticisms of the experts are now included.

  1. The order between the subsections “data analysis” and “Validity and reliability” was inverted. How data analysis of the open questions was done is explained in subsection 2.4, now in a more complete way.

  1. The data confidentiality is in fact a term more used for the limitations imposed to the data collected, especially to other persons outside the research team. Therefore, we describe the ethical principles more clearly.

  1. In tables, the commas were replaced by periods and the absolute frequencies were also included where they were missing. The “-“ were replaced by zero (0). We do not consider it is appropriate to reduce the subtitle size in each table, since it is better that the reader can understand the data/information presented correctly.

         To avoid misinterpretations “Don’t answer” was substituted by “Don’t justify.

The lines in the tables were suppressed and the lines are now confined to the header and lower limit of the tables

The wrong sum included in Table 4 was corrected.

The sum of each item is now exactly 100%.

The arguments associated with each diet continue to include only the absolute frequencies. Most of them have a low value and the inclusion of all percentages overloads the table with values. But, based on the percentage of each category, it is easy to infer the percentages of the arguments.

The data of Table 10 are now presented in another way and the mean excluded.

The information of Figures 1 and 2 is now included in a new Table (Table 11)

  1. The Site of the Vegetarian center includes the following information:

Em setembro de 2007, o Centro Vegetariano realizou o primeiro estudo a uma amostra representativa da população portuguesa para determinar o número de vegetarianos em Portugal, com a AC Nielsen.

Thus, it considers that the study involved a representative sample.

This information is used by different authors and published in different articles as well as in thesis on the subject. The main reason is certainly because is the only data available about this subject, which we confirmed after hours of research.

  1. The ideas presented in the article by Dhont & Hodson (2014) are now quoted exclusively on the main aim of their study.

  1. In the new version, the discussion section follows strictly the order of the objectives of the study. Explicit guidelines for teacher training of the present subject are included.

  1. One more limitation of the study has been included

Reviewer 2 Report

This study involved Portoguese pre-service teachers attending two higher education institutions, one in the urban context (UC) and the other in rural context (RC), whom have been administered a questionnaire. The aim of the study was assessing  partecipants' eating habits, perceptions and knowledge about the impact of meat consumption in human health, environment and animal welfare.

The results show that the participants of both groups had a preference for an omnivorous diet with no restrictions, considering this diet the best for human health but the worst for the environment and animal welfare.In addition, they also showed  a predisposition to reduce meat consumption for environmental and animal welfare reasons but they were often unable to properly justify their ideas and choices.

The manuscript is well structured but to make comprehension and reading easier a grammatical and linguistic revision of the entire text is necessary.

Below the list of the grammar mystakes and typographical errors found in the text and points to be clarified:

Line 27: please specify who is the author cited and add the reference to this sentence. Please check the first reference (the English translation of the title in particular)

Line 39: please rephrase the sentence in this way: “This openness of the Portuguese society also brought less positive aspects, such as the increase in consumption of fast food. In this respect, the first McDonald's restaurant opened in Lisbon in 1991..”

Line 43-44: please rephrase the sentence in this way“This trend  popular amongst university students appears as a consequence of timetables and academic tasks..”

Line 50: “(p.1) stands for ?

Line 57: please add “than the non-heme form” after “body”.

Lines 76-79: the AMR is fueled not only by veterinary prescriptions but also by the medical sector. Authors should discuss more this aspect referring also to veterinary controls carried out to ensure a proper use of drugs ( e.g. compliance with withdrawal times) to guarantee the consumer and, at the same time, the animal welfare achieving also a sustainable management of farms. In this regard, references could be improved citing:  Bozzo, G.; Corrente, M.; Testa, G.; Casalino, G.; Dimuccio, M.M.; Circella, E.; Brescia, N.; Barrasso, R.; Celentano, F.E. Animal Welfare, Health and the Fight against Climate Change: One Solution for Global Objectives. Agriculture 2021, 11, 1248. https://doi.org/10.3390/agriculture11121248

Lines 84-88: Please rephrase the sentence such as “More precisely, The World Health Organization (WHO) [18] claims, as some epidemiological studies suggest,  that processed meat (the one that is salted, smoked or dried) is carcinogenic and red meat is possibly carcinogenic. The WHO also advises that less than ten percent of  total energy consumption should originate in saturated fat, precisely the type present in different kinds of meat.” Add also references after “as some epidemiological studies suggest”.

Line 95: please replace “adjusted to” with “related to”

Line 96: please add the expression “The incidence of”  before “these last“

Line 113: please replace “of” with “in” before “biodiversity”

Line 115: please replace “of” with “in” before “freshwater”

Lines 126-128: Please rephrase in this way “through the belching due to enteric fermentation and in a smaller percentage through the intestine. Another main source of Methane emission is manure decomposition[28].”

Lines 318-319: The authors could provide more precise information on eating habits reported by the respondents.

Lines 320-321: The authors should explain better the result.  The use of “Accordingly” is not clear because firstly the partecipants selected the omnivore with no restrictions diet as the best diet and then as the worst.

Lines 407-408: Please rephrase the sentence in this way “Finally Table 10 evaluates how much the participants of each institution agree with the following three statements”

Line 525: Please correct the author’s name.

Line 561: Please replace “but if” with “although”

Lines 653-655: The authors should rephrase the sentence more clearly.

Lines 663-665: Please rephrase the sentence such as “That is why, it is considered that the topic deserves greater emphasis during teacher training courses, especially because the topic of food is part of the curriculum in the years of schooling for which these students are being prepared.”

The tables could be clearer. Table 3 shows next to the type of diet the number of answers. This information is missing in the other tables. Moreover, the answers shown in Table 5 are not the same as n.

The results obtained provide an advancement of the current knowledge on the actual state of school information and, especially, disinformation on animal welfare, climate change and the impact of meat consumption (and other human habits) on public health and environment. Data shows the need to improve teacher training and raise awareness among educators at all levels of education and training on these topics.

The conclusions are consistent with the evidence and arguments presented .

Taking all this into account the work fits the journal aims.

The manuscript is well structured but to make comprehension and reading easier a grammatical and linguistic revision of the entire text is necessary.

Author Response

Thanks for your comments and contributions for the improvement of the article. We really appreciate your suggestions.

Therefore, a few more references have been included.

Your English suggestions have also been included.

The article has been globally reviewed by an English native speaker

Reviewer 3 Report

The paper presented focuses on the study of a main social phenomenon whit great current interest: meat consumption and human and environmental health and animal welfare. This subject has a social and environmental impact due to the high meat consumption in most western countries. This research was focused on the region of Portugal and on the group of teachers. This approach is relevant, since it deals with a group that has great social influence.

The introduction is deep and exhaustive, showing the fundamental elements of the researched topic. It relates meat consumption with environmental impact, as well as in the studies developed by the WHO related to these behaviors. The introduction is clear and contextualizes the research well. Likewise, the objectives are also clear and reasonable.

The methodology is well explained and it is clearly shown how the research was conducted. In addition to using an adequate statistical study, an effort has also been made to validate the questionnaire used, as well as to verify the ethical aspects.

The results are well structured and defined. The data are well explained and provide relevant information. The tables are numerous and clear. The only problem I detected in this section is that Figures 1 and 2 could be improved a little if they were larger and the numbers could be read more clearly.The discussion is correct and well documented. In turn, the conclusions are clearly derived from the objectives and the results obtained. There is no doubt that this is a very good work of great interest to the scientific community.

Author Response

Thanks for your comments concerning the article.

Round 2

Reviewer 1 Report

Scientific Arbitration to the Article

The impact of meat consumption on human health, environment and animal welfare: perceptions and knowledge of pre-service teachers

Second round

1 – The efforts of the authors in significantly improving the content of the article are recognized.

However, it is considered that:

2 – The number and respective date of the Ethics Committee that gave a positive opinion to the study must be indicated.

3 – Table 2 must include the absolute and relative frequencies in all data. The presentation of relative frequency values must be consistent with regard to the number of places (in this case, decimal places) (eg., there is 8.7 in Omnivore with no restrictions, but in Omnivore with red meat restriction the value presented is 25, when it should be 25.0).

4 – The same nature of comments for table 3.

5 – The same nature of comments for table 4.

6 – The same nature of comments for table 5.

7 – The same nature of comments for table 6.

8 – The same nature of comments for table 7.

9 – The same nature of comments for table 8.

10 – The same nature of comments for table 9.

11 – There seems to be a formatting error after table 10. The graphics are unnecessary and should be removed. At most, some of your information can be written and analyzed in the narrative.

12 – Table 13 – adequate.

This is the opinion.

Author Response

To the reviewer

Concerning the new suggestions:

The reference of the Ethics Commission is included

All the percentages are included in every table.

The authors

Reviewer 2 Report

The authors have improved the manuscript as recommended.

Author Response

To the reviewer

No more suggestions have been presented.

The authors

Reviewer 3 Report

At first I thought the work was very good. Right now, I see that it has improved thanks to the review comments. 

Author Response

(The authors gave the same response as above.)
